# The Rqc2/Tae2 subunit of the ribosome-associated quality control (RQC) complex marks ribosome-stalled nascent polypeptide chains for aggregation

Ryo Yonashiro[1†], Erich B Tahara[1†‡], Mario H Bengtson[1§], Maria Khokhrina[2], Holger Lorenz[2], Kai-Chun Chen[1], Yu Kigoshi-Tansho[1], Jeffrey N Savas[3¶], John R Yates III[3], Steve A Kay[1], Elizabeth A Craig[4], Axel Mogk[2], Bernd Bukau[2], Claudio AP Joazeiro[1,5*]

[1]Department of Cell and Molecular Biology, The Scripps Research Institute, La Jolla, United States; [2]Zentrum für Molekulare Biologie der Universität Heidelberg (ZMBH), Deutsches Krebsforschungszentrum (DKFZ), DKFZ-ZMBH Alliance, Heidelberg, Germany; [3]Department of Chemical Physiology, The Scripps Research Institute, La Jolla, United States; [4]Department of Biochemistry, University of Wisconsin - Madison, Madison, United States; [5]Zentrum für Molekulare Biologie der Universität Heidelberg (ZMBH), Deutsches Krebsforschungszentrum-ZMBH Alliance, Heidelberg, Germany

*For correspondence: joazeiro@scripps.edu

†These authors contributed equally to this work

Present address: ‡Departamento de Bioquímica e Imunologia, Universidade Federal de Minas Gerais, Belo Horizonte, MG, Brazil; §University of Campinas, São Paulo, Brazil; ¶Department of Neurology, Feinberg School of Medicine, Northwestern University, Chicago, United States

Competing interests: The authors declare that no competing interests exist.

**Abstract** Ribosome stalling during translation can potentially be harmful, and is surveyed by a conserved quality control pathway that targets the associated mRNA and nascent polypeptide chain (NC). In this pathway, the ribosome-associated quality control (RQC) complex promotes the ubiquitylation and degradation of NCs remaining stalled in the 60S subunit. NC stalling is recognized by the Rqc2/Tae2 RQC subunit, which also stabilizes binding of the E3 ligase, Listerin/Ltn1. Additionally, Rqc2 modifies stalled NCs with a carboxy-terminal, Ala- and Thr-containing extension—the 'CAT tail'. However, the function of CAT tails and fate of CAT tail-modified ('CATylated') NCs has remained unknown. Here we show that CATylation mediates formation of detergent-insoluble NC aggregates. CATylation and aggregation of NCs could be observed either by inactivating Ltn1 or by analyzing NCs with limited ubiquitylation potential, suggesting that inefficient targeting by Ltn1 favors the Rqc2-mediated reaction. These findings uncover a translational stalling-dependent protein aggregation mechanism, and provide evidence that proteins can become specifically marked for aggregation.

## Introduction

Under various circumstances, translating ribosomes can halt NC elongation and become stalled, such as upon translation of mRNA templates lacking stop codons, containing sequential suboptimal codons, or encoding homopolymeric Lys tracts (*Wang et al., 2015*; *Comyn et al., 2014*; *Lykke-Andersen and Bennett, 2014*). Ribosome stalling poses a problem, as it can both reduce the pool of translation-competent ribosomes and give rise to aberrant—and potentially toxic—nascent poly-peptide chains (NCs). To prevent these undesirable consequences from taking place, stalled ribo-somes are rescued by factors that split the subunits, releasing the mRNA (for degradation by the exosome), the 40S subunit, and the 60S subunit stalled with a nascent peptidyl-tRNA conjugate, which is then targeted by the RQC complex (*Wang et al., 2015*; *Comyn et al., 2014*; *Lykke-*

**eLife digest** Cells use molecular machines called ribosomes to build proteins by connecting amino acids – the building blocks of proteins – together in a particular sequence. The chain of amino acids gradually lengthens as the protein forms, yet remains attached to the ribosome until the protein is complete.

While this process is underway, cells can check that a newly forming chain is not abnormal or damaged. If it is, a cell then essentially 'decides' on whether to correct or eliminate it. Such protein quality control processes are important for ensuring the health and fitness of cells and organisms. Recently, a new protein quality control mechanism was discovered that senses when a ribosome becomes jammed as it produces a new protein. This mechanism recycles the ribosome so it can make more new proteins. It also disposes of the stalled protein using a cell complex, called the ribosome-associated quality control complex, which is found in all eukaryotic organisms including yeast and humans. This protein complex consists of three subunits; one of which, called Rcq2, tags ribosome-stalled proteins with a "tail" that contains the amino acids alanine and threonine. However, the purpose of this tag was not clear.

Yonashiro, Tahara et al. now show that the tagging of ribosome-stalled proteins by Rqc2 in yeast cells induces the tagged proteins to clump together. This clumping probably prevents these proteins from inadvertently interfering with other molecules or processes within the cell. The formation of these clumps also correlates with the activation of a stress response in the cell, indicating that these clumps create a signal that prompts the cell to protect itself in response to the accumulation of more abnormal proteins.

Mutations in one subunit of the ribosome-associated quality control complex in mice cause a condition that resembles a neurological disease in humans, called amyotrophic lateral sclerosis or ALS for short. A future challenge is therefore to understand how much Rqc2-mediated tagging and clumping of ribosome-stalled protein has a role in this and other neurodegenerative diseases.

*Andersen and Bennett, 2014*). The RQC is minimally composed of the Ltn1 (Listerin in mammals), Rqc1, and Rqc2/Tae2 (NEMF in mammals) subunits (*Bengtson and Joazeiro, 2010*; *Brandman et al., 2012*; *Defenouillere et al., 2013*). How the RQC functions has only begun to be understood. According to the current model, Rqc2 first recognizes the stalled 60S and facilitates binding of the Ltn1 E3 ligase, which, in turn, ubiquitylates the aberrant NC (*Lyumkis et al., 2014*; *Shen et al., 2015*; *Shao et al., 2015*). Next, in a manner dependent on Rqc1, the Cdc48/VCP AAA ATPase and its ubiquitin-binding cofactors are recruited to the complex and facilitate NC delivery to the proteasome for degradation (*Brandman et al., 2012*; *Defenouillere et al., 2013*; *Verma et al., 2013*). It has been recently discovered that Rqc2 can, in addition, recruit Ala- or Thr-loaded tRNA to promote the C-terminal elongation of stalled NCs in a template- and 40S ribosomal subunit-independent manner (*Shen et al., 2015*). Such 'CAT tails' have no defined sequence and are heterogeneous in length, forming a smear extending as much as 5 kDa above the unmodified reporter band in SDS-PAGE. The physiological relevance of this process has remained unclear, however, as so far it has only been reported for Ltn1- or Rqc1-deficient cells (*Shen et al., 2015*). Moreover, the fate of CATylated NCs has remained unknown.

## Results

### Stalled translation can lead to the formation of nascent chain aggregates

We observed that, in *ltn1△* cells, in addition to the previously described increased steady-state levels and 'CATylation' of stalling reporters (*Bengtson and Joazeiro, 2010*; *Shen et al., 2015*), a fraction of those reporters migrated very slowly in gel electrophoresis, close to the loading well (*Figure 1A*; in the experiments presented, smears due to CATylation often run close to the unmodified reporter band, and can be more clearly observed with the smaller molecular weight reporters, PtnA-NS and GRR; see also below). The phenomenon was also observed with cells in which only

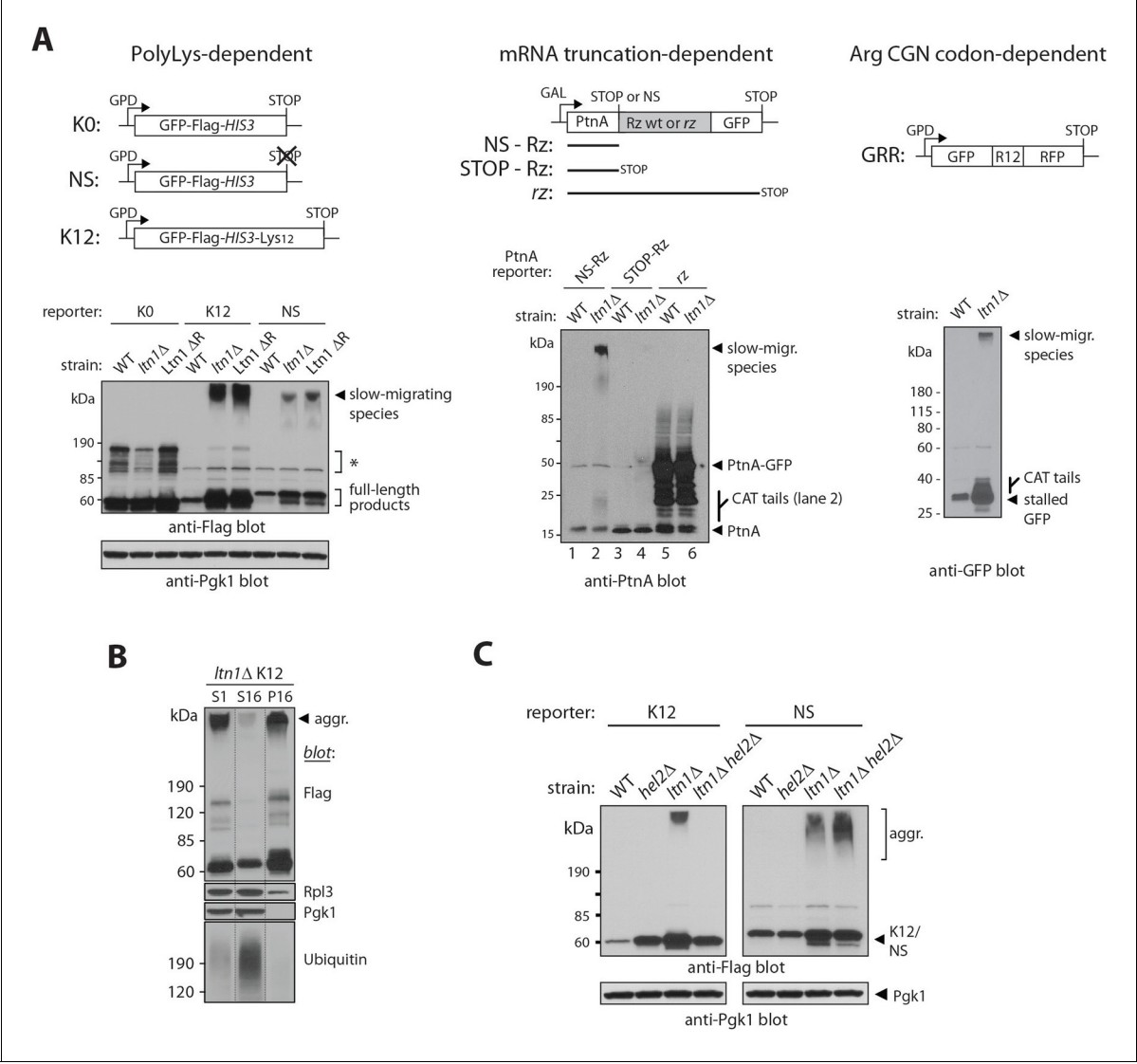

**Figure 1.** Stalled translation can lead to the formation of nascent chain aggregates. (**A**) *Top panels*, diagrams of reporter constructs encoding stalling-prone nascent chains and respective controls. PolyLys-dependent stalling (*left*): GFP-Flag-*HIS3* fusion protein control (K0), its *bona fide* nonstop (NS) protein derivative, and a derivative fused to 12 lysines (K12). Endonucleolytic mRNA cleavage-dependent stalling (*middle*): Protein A ZZ domain-Ribozyme-GFP fusion constructs. A self-cleaving ribozyme (Rz) within coding sequence generates a nonstop (NS) mRNA encoding stalled Protein A (PtnA). Controls are constructs with a cleavage-defective ribozyme generating a full-length PtnA-GFP fusion (*rz*), or with a stop codon preceding the Rz cleavage site (STOP-Rz), such that nascent PtnA is not expected to become stalled in ribosomes. Horizontal lines represent the encoded polypeptides. Arg CGN codon-dependent stalling (*right*): GFP-R12-RFP (GRR), where R12 is encoded by unpreferred Arg codons. *Lower panels*, reporter protein expression in a wild type strain (WT; BY4741), a *LTN1*-deleted strain (*ltn1△*) or a strain whose endogenous Ltn1 lacks the RING domain (Ltn1 △R). Immunoblots of SDS-boiled cell extracts: anti-Flag, anti-PtnA, or anti-GFP to monitor reporter expression, and anti-Pgk1 as loading control. The migration of CATylated species is indicated. Asterisks indicate bands of unknown identity. (**B**) Stalling reporter slow-migrating species are pelleted upon high speed centrifugation. The extract of a K12 reporter-expressing *ltn1△* strain was pre-cleared by centrifugation at 1000 x *g* for 5 min and its supernatant (S1) was then subjected to 16,000 x *g* for 10 min. The resulting supernatant (S16) and pellet (P16) were analyzed by western blot against Flag tag (K12), Rpl3 (a 60S ribosomal protein), Pgk1 (phosphoglycerate kinase 1, a soluble protein) and ubiquitin (high-molecular weight conjugates migrating above 120 kDa are shown). (**C**) Translational stalling is required for reporter aggregation. NS and K12 reporter protein expression in strains lacking Ltn1 and/or the translational stalling factor, Hel2.

The following figure supplement is available for figure 1:

**Figure supplement 1.** Stalled translation can lead to the formation of nascent chain aggregates.

Ltn1's E3-catalytic RING domain had been deleted (Ltn1 △R; *Figure 1A*, left panel) suggesting that it is prevented by Ltn1-mediated ubiquitylation under normal conditions.

We analyzed several stalling reporters that have been previously described. In one set, the reporters consist of a GFP-Flag-His3 fusion followed by a stop codon (K0), lacking stop codons (NS, for 'nonstop'), or followed by a 12 Lys track and stop codons (K12) (*Figure 1A*, left panel, and [*Bengtson and Joazeiro, 2010*; *Ito-Harashima et al., 2007*]). Translation of NS and K12 reporters is believed to stall due to polyLys tract synthesis. In a second set of reporters, the Protein A ZZ domain (PtnA) is followed by a stop codon (STOP-Rz), by a wild type self-cleaving ribozyme sequence (NS-Rz), or by a mutant ribozyme (*rz*) (*Figure 1A*, middle panel, and [*Wilson et al., 2007*]). In the case of the NS-Rz reporter mRNA, translating ribosomes stall as they reach the 3'end of the cleaved mRNA with no stop codons present; on the other hand, with the mutant *rz* reporter, the mRNA fails to be cleaved, allowing translation to proceed through an in-frame GFP sequence without stalling. In a third set of experiments, a GFP-12(Arg)-RFP fusion protein (GRR) was utilized (*Figure 1A*, right panel, and ref [*Ito-Harashima et al., 2007*]). In this case, stalling occurs as a result of the presence of multiple unpreferred Arg CGN codons (*Letzring et al., 2013*). Despite their unrelated encoded protein sequences and distinct stalling mechanisms, we were able to observe slow-migrating species for all stalling reporters examined, but not their respective parental controls (e.g., K0, STOP-Rz). The formation of slow-migrating reporter species thus appears to be translational stalling-dependent.

We next investigated the nature of these high-molecular weight species. Slow migration was not due to Ltn1-independent poly-ubiquitylation of stalling reporters, since migration was not shifted after treatment with the deubiquitylating enzyme, Usp2 (*Figure 1—figure supplement 1A*; [*Kaiser et al., 2011*]). We reasoned that those species might instead correspond to insoluble aggregates. Consistent with this possibility, slow-migrating reporter species were efficiently sedimented by centrifugation under conditions normally used to pellet protein aggregates (see, e.g., [*Fang et al., 2011*; *Koplin et al., 2010*]), in contrast to a soluble protein (Pgk1) or the bulk of high-molecular weight poly-ubiquitylated proteins in the extract (*Figure 1B*).

The ability to observe aggregates of stalling reporter proteins by western-blot implies that those aggregates are resistant to solubilization by boiling in 1% sodium dodecyl sulfate (SDS), as samples for the experiments above were subjected to this treatment prior to gel running. Resistance to ionic detergents is characteristic of ordered fibrillar structures such as the amyloid formed by yeast prions or by expanded polyglutamine (polyQ) tracts (e.g., [*Toyama and Weissman, 2011*; *Liebman and Chernoff, 2012*]). To our knowledge, this is the first report of E3 dysfunction leading to formation of aggregates sharing properties with amyloid.

However, in contrast to its effects on stalling reporters, deletion of *LTN1* failed to affect levels or stimulate aggregation of a Huntingtin exon 1 polyQ-GFP reporter carrying either a disease-associated expansion (Htt Q72) or a normal length tract (Htt Q25; *Figure 1—figure supplement 1B*; [*Krobitsch and Lindquist, 2000*]). Thus, loss of Ltn1 function is not associated with the increased formation of protein aggregates in general, but rather appears to be specifically associated with stalled NC aggregation.

To further verify that ribosome stalling is required for NC aggregation, we took advantage of the knowledge that the Hel2 protein is required for polybasic tract-mediated translational stalling (*Brandman et al., 2012*). Thus, in a *hel2△* background, ribosomes translating a 12-Lys tract in a stop codon-containing reporter (K12) would be expected to translate through the tract, reach the stop codon, and terminate translation normally, releasing the NC; on the other hand, the stalling of ribosomes translating a *bona fide* non-stop mRNA (e.g., NS) would not be expected to be prevented by *HEL2* deletion (*Brandman et al., 2012*). We thus asked what consequence *HEL2* deletion would have on reporter aggregation. As predicted, the results in *Figure 1C* show that *HEL2* deletion efficiently suppressed aggregation of K12—but not NS—in the *ltn1△* background.

## Rqc2-mediated modification of stalled nascent chains with CAT tails results in their aggregation

The formation of NC aggregates in Ltn1-deficient cells correlated with NC modification with CAT tails. Furthermore, the low-complexity CAT tail sequences (*Shen et al., 2015*) are reminiscent of aggregate-forming polyAla tracts (*Albrecht and Mundlos, 2005*). We thus hypothesized that the aggregation of stalled NCs was mediated by CAT tails.

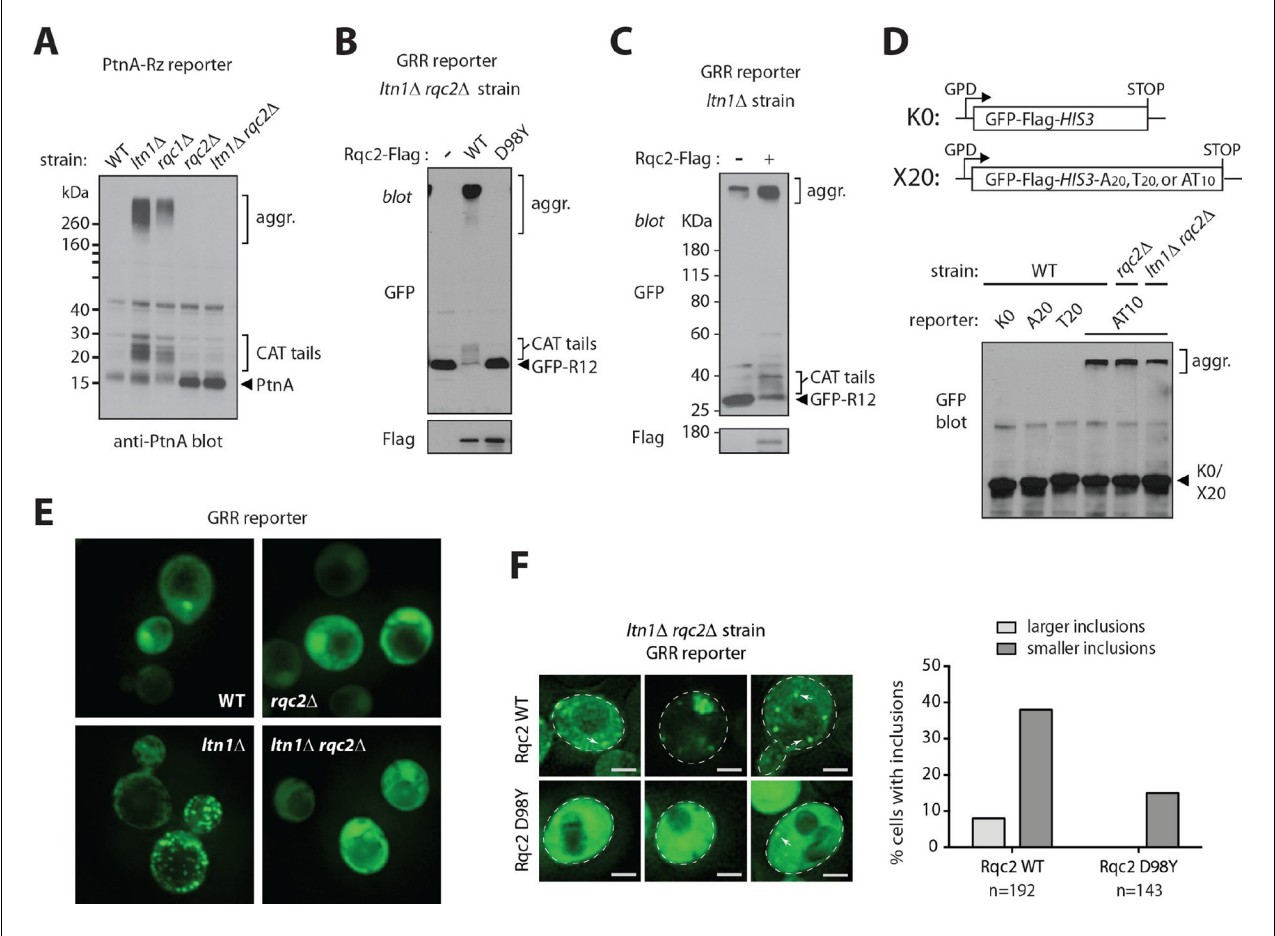

**Figure 2.** Rqc2-mediated modification of stalled nascent chains with CAT tails results in their aggregation. (**A**) NC CATylation correlates with aggregation—effects of *RQC1* and *RQC2* deletion. The indicated strains were transformed with the PtnA NS-Rz reporter. Reporter expression was monitored by immunoblot anti-PtnA. The migration of CATylated species is indicated. (**B**) An Rqc2 mutant defective in CAT tail synthesis fails to promote aggregation of stalled NCs. The *ltn1△ rqc2△* strain expressing the GRR reporter was transformed with plasmids encoding Rqc2-Flag wild type (WT) or D98Y mutant. (**C**) Endogenous Rqc2 is limiting for NC CATylation and aggregation in *ltn1△* cells. The *ltn1△* strain expressing the GRR reporter was transformed or not with plasmid encoding Rqc2-Flag wild type (WT). Reporter expression was monitored by immunoblot anti-GFP. (**D**) Fusion of a CAT tail-mimetic sequence to the C-terminus of the K0 reporter protein suffices to promote aggregation independently of stalling or Rqc2. *Top panel*, diagram of constructs. *Lower panel*, as in 'a'. The indicated strains were transformed with plasmids encoding the parental reporter (K0, as described in 1a) or its derivatives fused to a C-terminal tail of 20 Ala, 20 Thr, or 10 Ala-Thr repeats, as indicated. (**E**) *Punctae* formed by stalling reporters in intact cells correlate with aggregates observed in WCE. Fluorescence microscopy imaging of indicated strains expressing the GRR reporter. GFP-positive *punctae* can be observed in the *ltn1△* strain. (**F**) CAT tail-dependent incorporation of the GRR stalling reporter into *punctae*. *Left*, The *ltn1△ rqc2△* strain was transformed with plasmids encoding Rqc2-Flag wild type (WT) or D98Y mutant as in panel 'B' and examined by fluorescence microscopy. Three different distribution patterns of the GFP signal that are representative for each strain are shown. Arrows point to selected *punctae*. Scale bar, 2 μm. *Right*, Quantification of cells harboring GFP-positive inclusions in the *ltn1△ rqc2△* strains expressing Rqc2 WT or D98Y mutant.

The following figure supplement is available for figure 2:

**Figure supplement 1.** Rqc2-mediated modification of stalled nascent chains with CAT tails results in their aggregation.

Consistent with the hypothesis, stalled NC CATylation and aggregation were also observed in *rqc1△* cells (*Figure 2A*). The observation that *RQC1* deletion phenocopied *LTN1* deletion with regard to NC aggregation also implies that it is not a defect in Ltn1-mediated ubiquitylation per se—which is functional in the *rqc1△* background (*Brandman et al., 2012*; *Defenouillere et al., 2013*)—that causes aggregation (consistent with this interpretation, treatment of wild type yeast with the proteasome inhibitor MG132 led to stalling reporter accumulation without producing aggregates; *Figure 2—figure supplement 1A*). Rather, these results suggest that stalled NCs are

driven towards CATylation and aggregation as a result of a defect in a step downstream of ubiquity-lation but upstream of the proteasome, such as Cdc48/VCP recruitment (*Verma et al., 2013*).

Further correlation between CATylation and aggregation was obtained by inspecting the PtnA-Rz reporter in a Ltn1-deficient strain also lacking Rqc2—the results in *Figure 2A* show that, in the absence of CAT tails, NC aggregates also failed to form. We further examined the CAT tail require-ment for NC aggregation by using *ltn1△ rqc2△* strains expressing a stalling reporter and trans-formed with plasmids encoding either wild type Rqc2, or Rqc2 carrying a mutation in the highly conserved Asp98 residue. Asp98 is required for CAT tail synthesis but not for ribosome binding—in fact, the Rqc2 D98Y mutant is expressed normally and is fully competent to support Ltn1 function in *rqc2△* cells (*Figure 2—figure supplement 1B* and [*Shen et al., 2015*]). As predicted by the hypoth-esis, while overexpression of wild type Rqc2 led to the quantitative conversion of the stalling reporter into aggregated forms, the Rqc2 D98Y mutant was unable to promote stalling reporter aggregation (*Figure 2B*). Overexpression of wild type Rqc2 also led to more quantitative conversion of monomeric to CATylated and aggregated forms of the stalling reporter in *ltn1△* cells, suggesting that endogenous Rqc2 is limiting (*Figure 2C*). Moreover, differently from what we had observed for Hel2 requirement (*Figure 1C*), the CAT tail synthesis requirement for aggregation was evident with all stalling reporters examined, including NS (*Figure 2—figure supplement 1C*). Yet, this require-ment appeared to be specific to stalling reporters, as *RQC2* deletion did not affect polyQ reporter aggregation (*Figure 2—figure supplement 1D*).

The hypothesis also predicted that hard-coding a CAT tail on a stop codon-containing reporter construct might bypass the requirements for both stalling and Rqc2 for protein aggregation. To test this possibility, we generated a construct in which a tract of 10 Ala-Thr repeats [(AT)10] was fused to the C-terminus of the control reporter K0 with the intent to mimic a CAT tail, followed by stop codons. The (AT)10 reporter has no stalling sequence, so its encoded protein is not expected to be a target of the RQC. As shown in *Figure 2D*, (AT)10, but not homopolymeric constructs with 20 Ala (A20) or 20 Thr (T20), was indeed able to form aggregates. Given that homopolymeric Ala tracts have been previously implicated in amyloid formation (*Albrecht and Mundlos, 2005*), we presume that polyAla sequences may need to be longer in order to form aggregates under the conditions uti-lized here. Thus, the combination of Ala and Thr as found in CAT tails appears to be more prone to aggregate, which can be observed with a sequence as short as 20 amino acids long. These results suggest that, in addition to being required, CAT tails can also be sufficient for aggregate formation.

Arguing against the possibility of reporter aggregation being a post-lysis artifact, mixing *ltn1△* cells with K12-expressing *ltn1△ rqc2△* cells immediately prior to cell lysis did not support aggregate formation (*Figure 2—figure supplement 1E*). In order to obtain further evidence for the formation of NC aggregates in intact cells, we examined the distribution of the GRR stalling reporter under fluorescence microscopy (*Figure 2E*). In the WT strain transformed with GRR, the low GFP signal was evenly distributed throughout the cytoplasm, with 1–3 local accumulations of signal being observed; in contrast, *LTN1* deletion led to the appearance of numerous GFP *punctae*, characterized by foci with intense brightness over a low, diffuse cytosolic fluorescence background, in a third of cells [i.e., 15 out of 45 cells showing this phenotype; we note that aggregate formation in the *ltn1△* strain is limited by endogenous Rqc2 levels (see *Figure 2C* above)]. GFP *punctae* formation in *ltn1△* cells depended on Rqc2-mediated CATylation, as evidenced by the finding that the phenomenon could be suppressed by *RQC2* deletion (*Figure 2E*), and subsequently rescued by overexpression of Rqc2 wild type but not D98Y (*Figure 2F*). Thus, the appearance of GFP *punctae* in living cells corre-lated with the presence of protein aggregates in cell extracts.

The results presented so far indicate that NCs can be assembled into aggregates through a pro-cess triggered by ribosome stalling and requiring CAT tail synthesis by Rqc2. This implies that pro-teins can become 'tagged' for aggregate assembly, and that this happens while nascent chains are still associated with the 60S subunit.

## Sis1 association reveals endogenous stalled nascent chain aggregates

Because molecular chaperones are implicated in handling misfolded and aggregated proteins (e.g., [*Parsell et al., 1994*; *Mogk et al., 2015*; *Nillegoda et al., 2015*]) we next examined the association of candidate chaperones with stalling reporters. Consistent with previous reports indicating that yeast amyloid aggregates are typically bound to the Hsp40/J protein, Sis1 (e.g., [*Aron et al., 2007*; *Park et al., 2013*; *Yang et al., 2013*]), the binding of Sis1 to stalling reporters was readily apparent

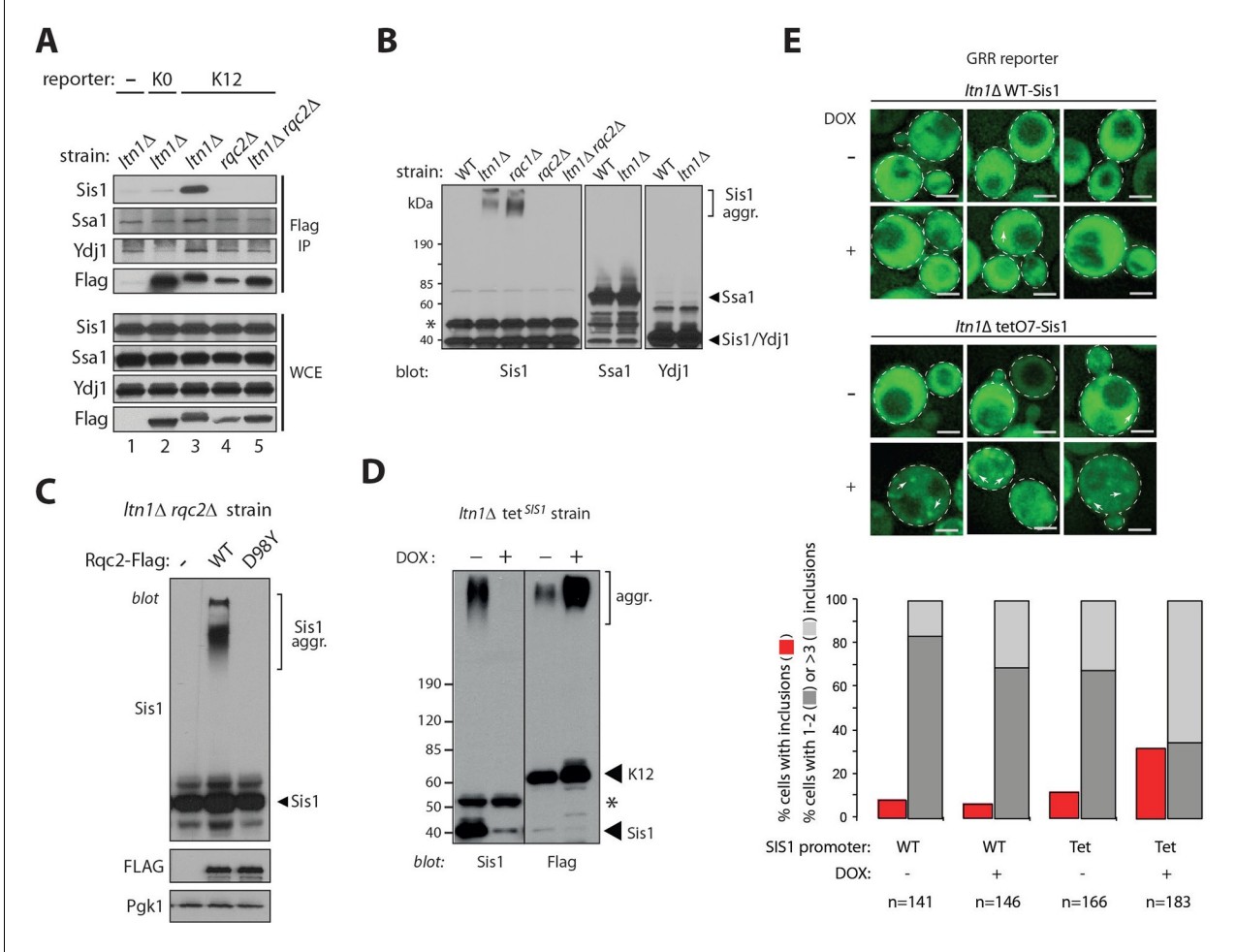

**Figure 3.** Sis1 association reveals endogenous stalled nascent chain aggregates. (**A**) Stalling reporters co-IP with Sis1 in an Rqc2-dependent manner. Whole cell extracts (WCE) of the indicated strains were Flag IP'ed (to pull down K0 or K12 reporters), followed by immunoblotting as indicated to the left of the panels. (**B**) Sis1 associates tightly with Rqc2-dependent aggregates formed by endogenous proteins in cells deficient for Ltn1 or Rqc1. WCE of the indicated strains were immunoblotted against Sis1, Ssa1, and Ydj1, as indicated. The ~47 kDa band in the Sis1 blot (asterisk) is nonspecific. (**C**) The formation of slow-migrating Sis1 species is dependent on Rqc2's ability to synthesize CAT tails. WCE of the *ltn1Δ rqc2Δ* strain expressing Rqc2-Flag wild type or D98Y mutant were analyzed by immunoblotting. (**D**) Sis1 depletion increases NC aggregation. *ltn1Δ* tetO7-Sis1 cells expressing the K12 stalling reporter were treated or not with doxycycline (DOX). WCE were analyzed by immunoblot against Sis1 (*left panel*) or Flag (*right panel*; for K12 detection). *Asterisk*, cross-reacting band. (**E**) Sis1 depletion increases GFP *punctae* formation in *ltn1Δ* cells. *ltn1Δ* WT-Sis1 or *ltn1Δ* tetO7-Sis1 cells expressing the GRR stalling reporter were grown for 24 hr in the presence (+) or absence (-) of doxycycline (DOX). *Top*, Three representative images are presented for each strain and treatment condition. Scale bar, 2 µm. Arrows point to selected *punctae*. *Bottom*, Quantification of cells harboring GFP *punctae* is represented by red bars; among those, the fraction of cells with 1 or 2 *punctae* is represented in dark gray, and the fraction of cells with 3 or more *punctae*, in light gray.

The following figure supplements are available for figure 3:

**Figure supplement 1.** Sis1 association reveals endogenous stalled nascent chain aggregates.

**Figure supplement 2.** *tetO7* promoter-dependent Sis1 depletion.

in extracts of a *ltn1△* strain (***Figure 3A*** and ***Figure 3—figure supplement 1A***). Other chaperones, such as Hsp70 Ssa1 and the Hsp40/J-protein Ydj1, also co-immunoprecipitated (co-IP'ed) with stalling reporters above background level but the differences were less marked and appeared more variable. Consistent with these observations, analysis of chaperones co-IP'ed with the K12 stalling reporter expressed in a *ltn1△* strain by mass spectrometry uncovered Sis1 among the most

abundant hits, and with the apparent highest signal-to-noise ratio [*Supplementary file 1* (Table SI)]. In contrast to the conspicuous co-IP of Sis1 with stalling reporters in *ltn1△* cells, markedly less Sis1 was pulled down by anti-Flag antibody-conjugated beads from a *ltn1△* strain expressing no Flag-tagged stalling reporters, from a *ltn1△* strain expressing the K0 parental reporter, or from *rqc2△* or *ltn1△ rqc2△* strains (*Figure 3A*). Thus, the Sis1 co-IP depended on stalling and on Rqc2, suggesting it binds stalled NCs *via* the CAT tails and/or aggregates.

Strikingly, a fraction of Sis1 itself exhibited slow-migrating species that were resistant to boiling in 1% SDS in *ltn1△* or *rqc1△* cells, but not in the *ltn1△ rqc2△* strain (*Figure 3B*). The specificity of this phenomenon is underscored by the failure to observe similar slow migrating species for Ssa1 or Ydj1 (*Figure 3B*). Like stalled NC aggregates, those Sis1 species were unaffected by treatment with Usp2cc (*Figure 3—figure supplement 1B*). Moreover, slow-migrating Sis1 species could be pulled down along with an IP'ed stalling reporter (K12 Flag IP; *Figure 3—figure supplement 1C*), and their formation was dependent on Rqc2's ability to synthesize CAT tails (*Figure 3C*). Together, these results suggest that Sis1 tightly associates with stalled NC aggregates. Importantly, given that the formation of these Sis1 species was independent of ectopic expression of stalling reporters, these findings also provide evidence for aggregate formation by endogenous stalled NCs, and suggest that stalling reporter aggregation is not an artifact caused by their overexpression.

The above observations raised the question of whether Sis1 plays a role in NC aggregation. Given that *SIS1* is an essential gene, to shed light onto this issue, stalling reporter aggregation was examined in a *ltn1△* strain in which *SIS1* expression is under the tetO7 promoter, so it can be turned off by treatment with doxycycline (*Aron et al., 2007*). The results in *Figure 3D* show that the levels of Sis1 in both free and aggregated forms were indeed reduced in response to doxycycline (left panel), and that this was accompanied by a marked increase in steady-state levels of NC aggregates (right panel). We also examined the effect of doxycycline in intact cells expressing the GRR stalling reporter (*Figure 3E*). Consistent with the immunoblot data in *Figure 3D*, the imaging results show that both the fraction of cells with GFP *punctae* and the number of *punctae* per cell were increased in *ltn1△* Tet-Sis1 cells (*Figure 3E*). In contrast, such effects were modest at best in *ltn1△* cells in which Sis1 expression is under its endogenous promoter and unaffected by doxycycline treatment (*Figure 3E* and *Figure 3—figure supplement 2*).

## Evidence for stalled nascent chain modification with CAT tails and aggregation in wild type cells

NC CATylation has so far only been observed with mutant yeast strains harboring inactivating mutations in Ltn1 or Rqc1, raising the issue of physiological relevance (*Shen et al., 2015*). This observation suggested it might also be possible to observe Rqc2-dependent effects in wild type cells by utilizing stalling reporters expected to be less easily targeted by Ltn1—e.g., with fewer potential ubiquitylation sites. To test this possibility, we generated reporters with all lysine residues mutated to arginine ('K-less') and fused or not to the R12 stalling sequence.

One set of reporters was based on HA-tagged GFP K-less fused or not to 12 suboptimal Arg codons that cause ribosomal stalling (K-less R12). Cells transformed with GFP K-less or GFP K-less R12 constructs were analyzed for HA tag expression (*Figure 4A*). Remarkably, CATylation of the GFP K-less R12 reporter was readily evident in the wild type strain, as indicated by the presence of a smear immediately above the monomeric reporter band (lane 3), which was dependent both on Rqc2 (compare lanes 3 and 5) and on the R12 stalling sequence (compare lanes 2 and 3).

However, under the above conditions, the formation of aggregates with the GFP K-less R12 reporter was not observed (*Figure 4A*). We presume this could be due to a detection issue, since GFP K-less was expressed at lower steady-state levels compared to GFP, perhaps due to imperfect folding (*Sokalingam et al., 2012*) [see shorter exposure; indeed, Rqc2- and stalling-dependent K-less reporter aggregates became evident in wild type cells treated with proteasome inhibitor and after concentrating the samples by affinity purification (*Figure 4—figure supplement 1*)].

The formation of stalling reporter aggregates in wild type cells was more conspicuous when analyzing a different set of reporters, based on Protein A ZZ domain-Rz construct (described in *Figure 1A*). Wild type cells transformed with stop codon-containing controls or Rz-dependent stalling constructs were analyzed for PtnA expression (*Figure 4B*). Regardless of whether or not they contained lysines, PtnA-STOP-Rz reporters exhibited a similar expression pattern, consisting of a major band of the expected size of the PtnA ZZ domain (lanes 1 and 3). As observed in *Figure 1A*,

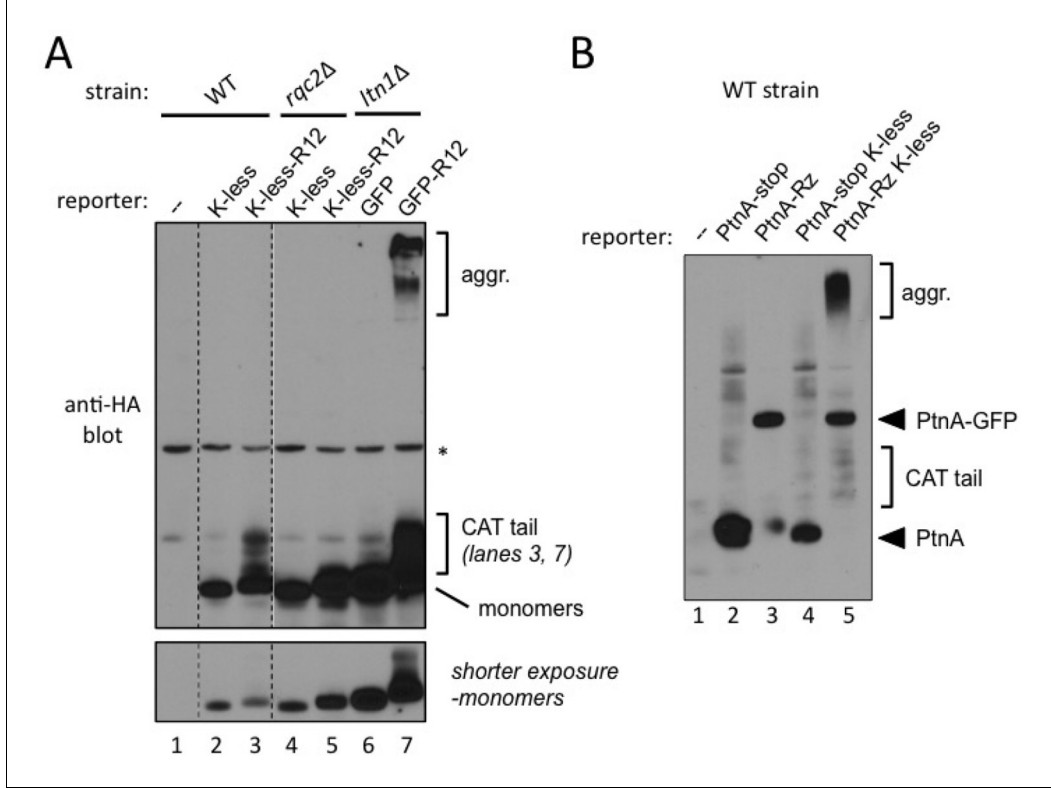

**Figure 4.** Evidence for stalled nascent chain modification with CAT tails and aggregation in wild type cells. (**A**) Stalling Lys-less reporter modification with CAT tails in wild type yeast. All constructs were HA-tagged. Expression of GFP, GFP-R12, GFP K-less ('K-less'), or GFP K-less R12 ('K-less-R12') reporter proteins in the indicated strains, revealed by anti-HA immunoblot. 'R12' is the stalling signal, consisting of 12 suboptimal Arg CGN codons. GFP-R12 expression in *ltn1△* cells is used as a control for aggregate formation. *Lower panel*, shorter exposure to reveal relative steady-state levels of monomeric reporter species. (**B**) Stalling PtnA-Rz reporter CATylation and aggregation in wild type yeast. All constructs were HA-tagged. Expression of PtnA-STOP-Rz, PtnA-Rz, PtnA-STOP-Rz K-less, and PtnA-Rz K-less reporter proteins in the wild type strain, revealed by anti-HA immunoblot.

The following figure supplement is available for figure 4:

**Figure supplement 1.** Stalling NC reporter aggregation in wild type yeast.

the PtnA-NS-Rz construct additionally encoded a protein corresponding in size to the product of full-length, uncleaved PtnA-(Rz)-GFP (lane 2). In contrast, in wild type cells expressing the PtnA-NS-Rz K-less construct, a band corresponding to the PtnA ZZ domain was not evident; instead, smears corresponding to its CATylated and aggregated forms were conspicuous (lane 4; compare to lane 2). We interpret this result as indicating that stalled NC encoded by the truncated PtnA-NS-Rz mRNA are normally targeted for degradation (compare lanes 1 and 2), but targeted for CATylation and aggregation if lysine ubiquitylation sites are not readily available. Furthermore, we conclude that Rqc2-dependent CATylation and aggregation of stalled NCs can both be observed in wild type cells with a functional RQC complex.

## Discussion

Stalled NC metabolism has recently emerged as a paradigm for the understanding of co-translational quality control, and includes mechanisms for handling aberrant NCs along with their encoding mRNA (*Wang et al., 2015*; *Comyn et al., 2014*; *Lykke-Andersen and Bennett, 2014*). In the context of the RQC complex, while Ltn1 mediates stalled NC ubiquitylation, Rqc2 mediates their CATylation (in addition to playing a non-essential role in supporting Ltn1 function). Rqc2 residues

implicated in CAT tail synthesis are conserved in evolution, suggesting an important function, but the fate of CAT tail-modified proteins had remained unknown. CATylation was not essential for Ltn1-mediated NC degradation (*Figure 2—figure supplement 1B*). Rather, here we report that CATylation promoted formation of NC aggregates—yet another process operating in stalled NC quality control. To our knowledge, this is the first demonstration that proteins can become specifically marked for aggregation.

Although physical characterization of NC aggregates remains to be carried out, evidence presented here indicate shared features with amyloid, including their dependency on a low-complexity aminoacid sequence bearing similarity to polyAla tracts, detergent insolubility, and binding to the J protein, Sis1. With regard to the latter, accumulating evidence suggests that Sis1 may function as an amyloid-recognition factor that enables Ssa- and Hsp104-mediated disassembly or random fragmentation (*Harris et al., 2014*). This proposed function provides a plausible explanation for our observation that Sis1 depletion led to increased NC aggregate levels, although it remains unclear whether and how regulation of presumably antagonistic Rqc2 and Sis1 activities controls steady-state levels of NC aggregates.

NC aggregation could have different functions, such as in sequestering aberrant NCs into forms less likely to interfere with cellular activities, or in mediating NC elimination *via* vacuolar degradation. Furthermore, NC aggregate formation correlates with the CATylation-dependent activation of Heat Shock Factor 1 (Hsf1) signaling observed in Ltn1-deficient cells (*Shen et al., 2015*). As protein aggregation has been previously shown to elicit Hsf1 activation (*Zou et al., 1998*), another conceivable role for NC aggregation is therefore in inducing translational stress signaling. On the other hand, excessive aggregate production certainly has the potential to cause toxicity. With these possibilities in mind, it will be important for future studies to investigate the role of NC aggregation in the neurodegenerative phenotype of Listerin-mutant mice (*Chu et al., 2009*).

## Materials and methods

### Reagents

Rabbit polyclonal antibodies were: anti-Protein A (Sigma, St. Louis, MI), anti-ubiquitin (Dako, Carpinteria, CA), anti-Sis1 (*Yan and Craig, 1999*), anti-Ydj1 (*Yan and Craig, 1999*), and anti-Ssa1 (*Lopez-Buesa et al., 1998*). Mouse monoclonal antibodies used were anti-Flag tag (M2; Sigma), anti-HA tag (12CA5; Roche, Germany), anti-Rpl3 (a gift of J. Warner), anti-GFP (Roche) and anti-Pgk1 (Invitrogen, Carlsbad, CA]). Secondary antibody was HRP-conjugated (Molecular Probes, Eugene, OR). Doxycycline hydrochloride was from Fischer Scientific (Waltham, MA). MG132 was from Cayman Chemical (Ann Arbor, MI). The recombinant catalytic core of Usp2cc was a generous gift of R. Kopito (*Kaiser et al., 2011*).

### *S. cerevisiae* strains

All strains used in this work are isogenic to BY4741 (MATa; *his3△1*; *leu2△0*; *met15△0*; *ura3△0*) or BY4742 (MATa; *his3△1*; *leu2△0*; *lys2△0*; *ura3△0*), except for the experiments shown in *Figure 1—figure supplement 1B*, in which the DS10 strain was used (MATa; *trp1△*; *lys1*; *lys2*; *ura3-52*; *leu2-3,112*; *his3-11,15*), and *Figures 3D and E*, in which the W303 strain was used (MATa; *leu2-3,112*; *trp1-1*; *can1-100*; *ura3-1*; *ade2-1*; *his3-11,15*). Derivative strains are shown in *Supplementary file 2* (Table SI). Mutant strains carrying single gene deletions were obtained commercially (Thermo Scientific, Waltham, MA). Additional deletions were performed by using cassettes designed to replace genes of interest with selection markers via homologous recombination. Primers used follow:

### *LTN1* deletion (using His3MX6 or KanMX6 cassettes)
FWD: 5′ TTGTTTAAAAAATGTAGTACATTTATATGAAATTTATATGCGATAGTCTAGAATTCGAGCTCGTTTAAAC;
 REV: 5′ AAATCTGCTAAGCCATCAAAAAAAGTTCAAGCAATAGTTGGTTCTTAATGCGGATCCCCGGGTTAATTAA

## Growth conditions

Cells were grown at 30° in SD-media. For Sis1 depletion *Sis1*-Tet Off strains were grown in SD-Trp media overnight, diluted in SD-Trp containing 10 µg/ml doxycycline to 0.05 $OD_{600}$ and grown for 12 hr. Cells were re-diluted in fresh SD-Trp containing doxycycline and incubated for further 12 hr to complete Sis1 deletion. It was necessary to keep the doxycycline incubation time as short as possible, as extended Sis1 depletion caused reduction in cellular growth and viability.

## Constructs

The Protein A constructs were a gift of A. van Hoof (UT-Houston) (*Wilson et al., 2007*). The Htt-polyQ constructs were gifts of S. Lindquist (Whitehead Institute) (*Alberti et al., 2009*). The GFP-R12-RFP (GRR) construct was a gift of O. Brandman (Stanford Univ.) (*Brandman et al., 2012*). The K0, NS, and K12 reporters have been described (*Bengtson and Joazeiro, 2010*; *Ito-Harashima et al., 2007*).

Constructs HA-GFP-stop-R12 ('GFP') and the stalling derivative, HA-GFP-R12-stop ('GFP-R12') were made on pRS316 (*URA3* marker; *CEN*) with the GPD promoter. These constructs were utilized as the basis for the GFP K-less derivatives: HA-GFP K-less-stop-R12 ('K-less') and HA-GFP K-less-R12-stop ('K-less-R12') by replacing Lys codons in the GFP coding sequence with the preferred Arg AGA or AGG codons through gene synthesis (IDT; sequence below). The R12 stalling sequence utilized unpreferred CGN Arg codons: CGG CGA CGA CGG CGC CGC CGG CGA CGA CGG CGC CGC. For Protein A K-less constructs, HA-tagged Protein A ZZ domain (wild type or K-less; stop codon-containing or non-stop) were generated by gene synthesis (IDT; sequence below) in the same way as the HA-GFP-K-less constructs above, and used to replace the homologous sequence upstream of the hammerhead ribozyme sequence of the PtnA-Rz construct gifted by A. van Hoof (UT-Houston).

### HA-GFP R12 - 789 bp

ATGTACCCATACGATGTTCCAGATTACGCTAGTAAAGGAGAAGAACTTTTCACTGGAGTTG
TCCCAATTCTTGTTGAATTAGATGGTGATGTTAATGGGCACAAATTTTCTGTCAGTGGAGAGGG
TGAAGGTGATGCAACATACGGAAAACTTACCCTTAAATTTATTTGCACTACTGGAAAACTACCTG
TTCCATGGCCAACACTTGTCACTACTCTGACGTATGGTGTTCAATGCTTTTCCCGTTATCCGGATCA
TATGAAACGGTATGACTTTTTCAAGAGTGCCATGCCCGAAGGTTATGTACAGGAACGCACTATA
TCTTTCAAAGATGACGGGAACTACAAGACGCGTGCTGAAGTCAAGTTTGAAGGTGATACCCTTG
TTAATCGTATCGAGT

 TAAAAGGTATTGATTTTAAAGAAGATGGAAACATTCTCGGACACAAACTCGAGTACAACTA
TAACTCACACAATGTATACATCACGGCAGACAAACAAAAGAATGGAATCAAAGCTAACTTCAAAA
TTCGCCACAACATTGAAGATGGATCCGTTCAACTAGCAGACCATTATCAACAAAATACTCCAA
TTGGCGATGGCCCTGTCCTTTTACCAGACAACCATTACCTGTCGACACAATCTGCCC
TTTTGAAAGATCCCAACGAAAGCGTGACCACATGGTCCTTCTTGAGTTTGTAACTGCTGC
TGGGATTACACATGGCATGGATGAACTATACAAAACTAGCCGGCGACGACGGCGCCGCCGGC-
GACGACGGCGCCGCtgatga

### HA-K-less GFP R12 - 789 bp

ATGTACCCATACGATGTTCCAGATTACGCTAGTAgAGGAGAAGAACTTTTCACTGGAGTTG
TCCCAATTCTTGTTGAATTAGATGGTGATGTTAATGGGCACAgATTTTCTGTCAGTGGAGAGGG
TGAAGGTGATGCAACATACGGAAggCTTACCCTTAggTTTATTTGCACTACTGGAAgACTACCTG
TTCCATGGCCAACACTTGTCACTACTCTGACGTATGGTGTTCAATGCTTTTCCCGTTATCCGGATCA
TATGAgACGGTATGACTTTTTTCAgaAGTGCCATGCCCGAAGGTTATGTACAGGAACGCACTATATC
TTTCAgAGATGACGGGAACTACAgaACGCGTGCTGAAGTCAgGTTTGAAGGTGATACCCTTGTTAA
TCGTATCGAGTTAAggGGTATTGATTTTAgAGAAGATGGAAACATTCTCGGACACAgACTCGAG
TACAACTATAACTCACACAATGTATACATCACGGCAGACAgACAAAgGAATGGAATCAgAGC-
TAACTTCAggATTCGCCACAACATTGAAGATGGATCCGTTCAACTAGCAGACCATTATCAACAAAA
TACTCCAATTGGCGATGGCCCTGTCCTTTTACCAGACAACCATTACCTGTCGACACAATCTGCCC
TTTTGAgAGATCCCAACGAAagGCGTGACCACATGGTCCTTCTTGAGTTTGTAACTGCTGCTGGGA
TTACACATGGCATGGATGAACTATACcgAACTAGCCGGCGACGACGGCGCCGCCGGCGAC-
GACGGCGCCGCtgatga

HA-Protein A ZZ domain - 492 bp
ATGTACCCATACGATGTTCCTGACTATGCGGGCTATCCCTATGACGTCCCGGACTATGCAGGA
TCCTATCCATATGACGTTCCAGATTACGCTCCGGCCGCCGCATGCCTTGCGCAACACGA
TGAAGCCGTAGAtAAtAAATTCAACAAAGAACAgCAAAAtGCGTTCTAcGAGATAtTTgCATTTgCC
TAACTTAAACGAAGAACAACGcAACGCaTTCATaCAAAGTTTgAAAGATGACCCtAGCCAAAGtGC-
cAAtCTaTTgGCtGAAGCcAAAAAGCTgAATGATGCaCAaGCaCCGAAAGTcGACAACAAATTCAA-
CAAAGAACAACAAAACGCGTTCTATGAGATCTTACATTTACCTAACTTAAACGAAGAACAGC-
GAAACGCCTTCATCCAAAGTTTAAAAGATGACCCAAGCCAAAGCGCTAACCTTTTAGCAGAAGC
TAAAAAGCTAAATGATGCTCAGgCGCCGAAAGTAGACGCGAATGGA

HA-K-less protein A ZZ domain - 492 bp
ATGTACCCATACGATGTTCCTGACTATGCGGGCTATCCCTATGACGTCCCGGACTATGCAGGA
TCCTATCCATATGACGTTCCAGATTACGCTCCGGCCGCCGCATGCCTTGCGCAACACGA
TGAAGCCGTAGAtAAtAgATTCAACAgAGAACAgCAAAAtGCGTTCTAcGAGATAtTTgCATTTgCC
TAACTTAAACGAAGAACAACGcAACGCaTTCATaCAAAGTTTgAgAGATGACCCtAGCCAAAGtGC-
cAAtCTaTTgGCtGAAGCcAgAAgaCTgAATGATGCaCAaGCaCCGAgAGTcGACAACAgATTCAACA-
gAGAACAACAAAACGCGTTCTATGAGATCTTACATTTACCTAACTTAAACGAAGAACAGC-
GAAACGCCTTCATCCAAAGTTTAAgAGATGACCCAAGCCAAAGCGCTAACCTTTTAGCAGAAGC
TAgAAgaCTAAATGATGCTCAGgCGCCGAgAGTAGACGCGAATGGA

For Rqc2 expression in yeast, the coding sequence of *S. cerevisiae RQC2* was amplified by PCR with the 3xFLAG epitope added at the C-terminus, and cloned into the YCplac111 vector (*LEU2* marker, *CEN*) with the GPD promoter. Mutants were constructed by site-directed mutagenesis using QuikChange Lightning kit (Stratagene, Santa Clara, CA).

## Protein expression analyses
MG132 treatment of yeast cells was as described (*Liu et al., 2007*). Total soluble extracts were prepared as described (*Bengtson and Joazeiro, 2010*). Protein quantitation was performed by the BCA method. 7.5–30 µg of protein extract were resuspended in 1% SDS, 0.005% bromophenol blue, 5% glycerol, 50 mM dithiothreitol, 50 mM Tris-Cl (pH 6.8) and incubated at 100°C for 5 min before fractionation by gel electrophoresis (4-20% Tris-Glycine gel, Invitrogen) and immunoblotting.

## Usp2-mediated deubiquitylation
30 µg of *ltn1△* cell extracts expressing K12 were treated with the recombinant catalytic core of Usp2 (Usp2cc; 5 µM) for 1h at room temperature, as described (*Kaiser et al., 2011*). The reaction was stopped by adding DTT-containing SDS-PAGE loading buffer and boiling for 3 min.

## Aggregate fractionation by high speed centrifugation
Aggregate fractionation was performed as described (*Fang et al., 2011*; *Koplin et al., 2010*) with minor modifications. Briefly, cells were grown to late logarithmic phase ($A_{600}$ = 0.8) and harvested by centrifugation at 2000 x *g* for 2 min. The cell pellet was resuspended in 750 µL cold non-denaturing lysis buffer [1% Triton, 140 mM NaCl, 1.5 mM $MgCl_2$, EDTA-free protease inhibitor cocktail (Roche), 20 U/mL RNase inhibitor (RNaseOUT, Invitrogen) and 10 mM Tris-Cl (pH 7.4)] and combined with 750 µL of 0.5-mm diameter glass beads. Suspensions were homogenized (three times for 45 s in a FastPrep FP120 Savant homogenizer) and lysates were transferred to clean tubes and centrifuged at 1000 x *g*, for 10 min at 4°C. Pre-cleared cell extract supernatants (S1) were further fractionated at 16,000 x *g* for another 10 min to pellet aggregates.

## Co-Immunoprecipitation
Cells were grown to late logarithmic phase ($A_{600}$ = 0.8) and harvested by centrifugation at 2000 x *g* for 2 min. The cell pellet was resuspended in 750 µL of cold non-denaturing lysis buffer [0.2% NP40, 140 mM NaCl, 1.5 mM $MgCl_2$, 1x EDTA-free protease inhibitor cocktail (Roche), 20 U/mL of RNase inhibitor (RNaseOUT; Invitrogen) and 10 mM Tris-Cl (pH 7.4)] and combined with 750 µL of 0.5-mm diameter glass beads. Suspensions were homogenized (three times for 45 s in a FastPrep FP120 Savant homogenizer) and lysates were transferred to clean tubes and centrifuged at 2800 x *g*, for 10 min at 4°C. 1 mg of soluble extract was incubated with 1 µg of anti-Flag antibody overnight at

4°C. Next, 600 µg of washed protein G magnetic beads (Life Technologies) were added and incubated for 2 hr at 4°C. Beads were pelleted and washed 5 times with cold lysis buffer before proteins were eluted by boiling in sample buffer.

## Immunoprecipitation of K-less aggregates

For the experiment presented in *Figure 4—figure supplement 1*, Cells were grown in SD media lacking uracyl, supplemented with 0.1% proline and 0.003% SDS until OD600 = 0.8, followed by treatment with MG132 (75 µM) or DMSO for 2 hr. Cells were lysed with 2% SDS lysis buffer and extracts were boiled for 3 min. Samples were diluted 20-fold in 0.5% triton lysis buffer and IP'ed with HA-agarose (Sigma) for 3 hr at 4C before running on SDS-PAGE.

## Imaging

For image acquisition and analysis, cells were grown as indicated and harvested by centrifugation and resuspended in PBS. Widefield microscopy was performed with an Olympus xcellence IX81microscope system using a 100x/1.45 NA Plan-Apochromat oil objective lens (Olympus, Japan) and a single band GFP filter set (AHF, Germany). As fluorescence light source the illumination system MT20 (Olympus) with a 150 W Xe arc burner was used. Z-stacks of images were recorded with slice distances of 200 nm and displayed as maximum intensity projections. Deconvolution of widefield images from Z-stacks was performed by using the Wiener filter of the Olympus xcellence software. Image processing for final figure preparation was performed with ImageJ (*Schneider et al., 2012*). For quantification of phenotypes, several areas with cells were randomly acquired and 45 or more cells were used for the analysis.

## Multidimensional protein identification technology (MudPIT) and LTQ mass spectrometry

Exponentially growing cells were collected, washed in cold water and lysed using glass beads in 10 mM Tris-HCl pH 7.4, 140 mM NaCl, 1.5 mM $MgCl_2$ and 0.2% NP40. 40 mg of total protein were used for IP with 40 µg Flag antibody and Protein A Dynabeads for 6 hr at 4°C. After 4 washes in lysis buffer, proteins were eluted in 200 µl of 300 µg/ml 3X Flag peptide for 30 min, at room temperature. Eluted samples were TCA-precipitated, resuspended in 8 M urea, and treated with ProteasMAX (Promega, Madison, WI) per the manufacturer's instruction. Subsequently, samples were reduced by 20 min incubation with 5 mM TCEP (*tris*(2 carboxyethyl)phosphine) at room temperature, alkylated in the dark by treatment with 10mM Iodoacetamide for 20 min, and quenched with excess TCEP. Proteins were digested overnight at 37 degrees with Sequencing Grade Modified Trypsin (Promega) and the reaction was stopped with formic acid.

The protein digest was subjected to cation-exchange microcapillary chromatography, and elutates from the column were electrosprayed directly into an LTQ mass spectrometer (ThermoFinnigan, Palo Alto, CA). A cycle of one full-scan mass spectrum (400–2000 m/z) followed by 7 data-dependent MS/MS spectra at a 35% normalized collision energy was repeated continuously throughout each step of the multidimensional separation. Application of mass spectrometer scan functions and HPLC solvent gradients were controlled by the Xcalibur datasystem.

Protein identification and quantification analysis were done with Integrated Proteomics Pipeline (IP2, Integrated Proteomics Applications, Inc. San Diego, CA). Tandem mass spectra were searched against the *Saccharomyces* Genome Database (SGD) protein database (http://www.yeastgenome.org/download-data/sequence, released on 01-05-2010). LTQ data was searched with 3000.0 milliamu precursor tolerance and the fragment ions were restricted to a 600.0 ppm tolerance. The ProLuCID search results were assembled and filtered using the DTASelect program (version 2.0) (*Cociorva, 2007*; *Tabb et al., 2002*) with false discovery rate (FDR) of 0.15, under such filtering conditions, the estimated false discovery rate was less than 5.4% at the protein level in each analysis.

## Acknowledgements

We thank J Warner, A van Hoof, R Kopito, O Brandman, and S Lindquist for reagents. EBT gratefully acknowledges the Brazilian Council for Scientific and Technological Development (CNPq) for a Postdoctoral Fellowship. MK was supported by the Hartmut Hoffmann-Berling International Graduate School of Molecular and Cellular Biology (HBIGS). JNS gratefully acknowledges the NRSA for Post-

doctoral Fellowship F32AG039127. Work in the Joazeiro laboratory was supported by NIH R01 grants NS075719 from the National Institute of Neurological Disorders and Stroke (NINDS) and CA152103 from the National Cancer Institute (NCI). Work in the Yates laboratory was supported by NIH grants P41 GM103533 and R01 MH067880. Work in the Craig laboratory was supported by NIH R01 grant GM31107 from the National Institute of General Medical Sciences (NIGMS). Work in the Bukau and Joazeiro (ZMBH) laboratories was supported in part by a grant of the Deutsche Forschungsgemeinschaft (SFB1036). This is manuscript 29225 from The Scripps Research Institute.

## Additional information

### Funding

| Funder | Grant reference number | Author |
| --- | --- | --- |
| Conselho Nacional de Desenvolvimento Científico e Tecnológico | 202144/2011-9 | Erich B Tahara |
| National Institutes of Health | F32AG039127 | Jeffrey N Savas |
| National Institutes of Health | GM103533 | John R Yates III |
| National Institutes of Health | MH067880 | John R Yates III |
| National Institutes of Health | GM31107 | Elizabeth A Craig |
| Deutsche Forschungsgemeinschaft | SFB1036 | Axel Mogk Bernd Bukau Claudio AP Joazeiro |
| National Institutes of Health | NS075719 | Claudio AP Joazeiro |
| National Institutes of Health | CA152103 | Claudio AP Joazeiro |

The funders had no role in study design, data collection and interpretation, or the decision to submit the work for publication.

### Author contributions

RY, EBT, MHB, MK, JNS, AM, Conception and design, Acquisition of data, Analysis and interpretation of data, Drafting or revising the article; HL, Acquisition of data, Analysis and interpretation of data, Drafting or revising the article; K-CC, Acquisition of data, Analysis and interpretation of data; YK-T, Conception and design, Acquisition of data, Analysis and interpretation of data; JRY, SAK, Analysis and interpretation of data, Drafting or revising the article; EAC, Conception and design, Analysis and interpretation of data, Drafting or revising the article, Contributed unpublished essential data or reagents; BB, CAPJ, Conception and design, Analysis and interpretation of data, Drafting or revising the article

### Author ORCIDs

Claudio AP Joazeiro, http://orcid.org/0000-0003-1433-8013

## Additional files

### Supplementary files

• Supplementary file 1. Mass spectrometry analysis of chaperones co-IP'ed with K0 (stop codon-containing control) and K12 (stalling) reporters from the *ltn1△* strain. K0 and K12 reporters were Flag IP'ed from WCE and analyzed by mass spectrometry. Peptide counts of co-IP'ed chaperones are shown. As represented in diagrams in *Figure 1A*, the K0 and K12 reporters consist of a GFP-Flag-*HIS3* backbone. Thus, *HIS3* peptide counts, which were the most abundant in the analyses, are presumed to be derived from the IP'ed reporter. We note that these analyses have limited ability to distinguish among homologous proteins with high degree of similarity, such as Ssa1/Ssa2, Hsc82/Hsp82, and Ssb1/Ssb2.

• Supplementary file 2. Strain genotypes.

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
