## [Decision Letter]

Thank you for submitting your work entitled "Rqc2/Tae2 marks ribosome-stalled nascent polypeptide chains for aggregation" for consideration by *eLife*. Your article has been favorably evaluated by Tony Hunter (Senior editor) and three reviewers, one of whom is a member of our Board of Reviewing Editors.

The reviewers have discussed the reviews with one another and the Reviewing editor has drafted this decision to help you prepare a revised submission.

Summary:

The authors describe the role of a ribosome quality control subunit Rqc2 in recruitment of alanine- and threonine-charged tRNAs to trigger the tagging of NCs with C-terminal Ala/Thr extensions (CAT tails). CAT tails foster the formation of SDS-resistant protein aggregates. This process is favored when ubiquitylation is inhibited by inactivation of Ltn1 or Rcq1. Importantly, they show aggregate formation of endogenous stalled NCs and provide evidence that CATylation and aggregation does also occur in wild-type cells. The manuscript is nicely written, the experiments are technically sound and support the authors’ conclusions. Altogether the work reveals an important novel aspect of ribosome-associated quality control.

Essential revisions:

Following discussion among reviewers it was agreed that two major points should be addressed to strengthen the role of aggregation and the chaperone Sis1 in this process.

1) The presence of Sis1 in the aggregates is intriguing. What happens to the RQC aggregate formation in cells lacking Sis1?

2) It would be nice to provide an explanation why the amount of free chain was not increased in the absence of aggregates – addition of poly-Lys (Figure 1) and depleting Rqc2 (Figure 2) abolished the aggregates formation. But why the free products (NS/K12) were not increased in a corresponding manner?

*Reviewer #1:*

The authors study ribosome-associated quality control complex (RQC) that promotes the proteasomal degradation of ribosome-stalled nascent chains. It has been recently described that the lack of E3 ligase Ltn1, a subunit of ROC Rqc2 makes nascent chains modified with C-terminal Ala- and Thr-containing tails ("CAT tails").

In this manuscript the authors nicely demonstrate that these CAT tails mediate the formation of amyloid-like nascent chain aggregates. It was proposed that Rqc2 acts as a CAT tail synthase that marks ribosome-stalled nascent chains for aggregation. CAT tails promote formation of insoluble nascent chain aggregates. Formation of NC aggregates was promoted after inactivating Ltn1 or by analyzing NCs with limited ubiquitylation potential, indicating that deficiency in Ltn1 promotes Rqc2-mediated NC CATylation and aggregation.

*Reviewer #2:*

In this manuscript Yonashiro et al. investigate the role of the so-called CAT tail that is added to nascent polypeptide chains (NCs) on stalled 60S ribosomes. When ribosomes become stalled a ribosome quality control complex (RQC) comprising, Rqc1 and Rqc2 and the ubiquitin ligase Ltn1 is recruited to the 60S subunit. While Ltn1/Rcq1 are critical for the degradation of nascent chains by the UPS, Rqc2 was recently shown to recruit alanine- and threonine-charged tRNAs to trigger the tagging of NCs with C-terminal Alanine and Threonine extensions (CAT tail). So far, however, the fate of CATylated NCs has remained enigmatic.

Here, the authors provide evidence that CAT tails foster the formation of SDS resistant protein aggregates. Using different reporters they demonstrate translational-stalling dependent CATylation and aggregate formation. Moreover, they show that this process is mediated by Rqc2 and is favored when ubiquitylation is inhibited by inactivation of Ltn1 or Rcq1. Importantly, they show aggregate formation of endogenous stalled NCs and provide evidence that CATylation and aggregation does also occur in wild-type cells.

The manuscript is clearly written, the experiments are technically sound and do support the authors’ conclusions. Altogether the work reveals an important novel aspect of ribosome-associated quality control. Even though the function of NC aggregates remains to be fully elucidated, it could for example be linked to the observed neurodegeneration in Listerin-mutant mice as discussed by the authors. Given the novelty, significance and quality of the work I therefore do recommend its publication in *eLife*.

*Reviewer #3:*

Ribosome-associated quality control (RQC) is an emerging topic with lots of surprises to be discovered. In this study, Yonashiro et al. reported that Rqc2 not only mediates the tagging of nascent chains with AT sequences, but also promotes their aggregation if the tagged nascent chains are not ubiquitinated or degraded. While the former finding was reported by Shen et al. (Science 347, 2015), the aggregation feature reported by this study is interesting and significant. In particular, it might shed light on the neurodegenerative phenotypes of mice lacking Ltn1. The enthusiasm about the aggregation feature of RQC is reduced a bit by some incomplete results interpretation. The role of chaperone Sis1 in this process, although interesting, remains obscure. With more supporting evidence from additional experiments, I would support publication of this exciting story in *eLife*.

Major concerns:

1) The authors nicely used many different approaches to prevent high molecular weight aggregates formation. For instance, addition of poly-Lys (Figure 1) and depleting Rqc2 (Figure 2) abolished the aggregates formation. But why the free products (NS/K12) were not increased in a corresponding manner? Where were the non-aggregated products?

2) In Figure 2, cells lacking Ltn1 apparently had more aggregates then cells lacking Rqc1. Does this mean ubiquitination could play a role in reducing aggregates formation? Is it possible to show the ubiquitin levels of the reporter (NS/K12) in these two strains?

3) In Figure 2, it is clear that only the sequence of AT10 is able to trigger the aggregation of the GFP reporter. But in WT cells, AT10-tagged GFP is evidently resistant to degradation. Is there any explanation about this discrepancy?

4) The authors provide strong evidence that aggregates of the nascent chain is due to the "CAT" tailing. But it is still suggestive at most. It would be ideal to demonstrate that the aggregates indeed contain the unique AT sequence. Mass spectrometry might be possible in this case.

5) The presence of Sis1 in the aggregates is intriguing. But what is the role of Sis1 in this process? Sis1 has been reported to influence prion formation (Harris et al. PLoS Genet 2014). What happens to the RQC aggregate formation in cells lacking Sis1?

6) In Figure 4, the authors pointed out K-less R12 reporter could be modified but failed to form aggregates. Is Sis1 still interacting with these species? This is important information that could explain the different behavior of this reporter in wild type cells.

---

## [Author Response]

Essential revisions:

*Following discussion among reviewers it was agreed that two major points should be addressed to strengthen the role of aggregation and the chaperone Sis1 in this process.*

*1) The presence of Sis1 in the aggregates is intriguing. What happens to the RQC aggregate formation in cells lacking Sis1?*

This is an interesting question that we have addressed experimentally with successful results. Given that *SIS1* is an essential gene in *S. cerevisiae*, in order to address the issue we have had to generate a Ltn1-deficient strain with *SIS1* expression under the control of a regulatable promoter (Tet-repressible, in our case). We show, using both biochemical and imaging data, that the *ltn1*∆ *tetO7*-Sis1 strain formed nascent chain aggregates under normal growth conditions, but that aggregate steady-state levels were greatly increased upon repression of Sis1 expression (Figure 3). This finding is consistent with mounting evidence indicating that Sis1 enables the Hsp70 Ssa and Hsp104 to fragment or disassemble amyloid aggregates.

*2) It would be nice to provide an explanation why the amount of free chain was not increased in the absence of aggregates – addition of poly-Lys (Figure 1) and depleting Rqc2 (Figure 2) abolished the aggregates formation. But why the free products (NS/K12) were not increased in a corresponding manner?*

The reviewers’ statement addresses results obtained from two very distinct experimental conditions – the effect of poly-Lys addition to cell extracts (former Figure 1) and the in vivo effect of genetically depleting Rqc2 (former Figure 2) – so we discuss those results separately below.

Regarding the poly-Lys issue:

It is indeed the case that, even though poly-Lys addition to cell extracts led to a marked decrease in aggregate signal, the amount of monomeric nascent chains was not obviously increased under these conditions as might have been expected. One possibility is that in aggregate levels in the extracts were very low relative to monomeric protein, but that the antibodies used recognize the epitope better in the aggregate form (e.g., due either to avidity or to “masking” proteins being less abundant in the higher molecular weight range). Another possibility is that poly-Lys addition leads to fragmentation of aggregates; the signal from those fragments would be spread over a wide range of molecular weights in the gel lane, thus making detection difficult. Finally, it is possible that aggregates precipitated with the addition of poly-Lys, although we do not think this is likely because we were attentive to vortex and load the entire sample into the gels.

We tested these and other possibilities experimentally, but have had limited success in generating conclusive data one way or the other. We have therefore decided to remove the experiments involving poly-Lys altogether, since the point made by those results was not central to our findings.

Regarding the Rqc2 deletion issue:

For some of the results reported, including those in the former Figure 2, reporter aggregate levels appear to be low compared to reporter monomer levels, so a detectable increase in the latter would not necessarily have been expected as a consequence of aggregates failing to form. Furthermore, we present more clear-cut results involving Rqc2 wt or D98Y overexpression over the *ltn1rqc2Δ*background (Figure 2); by comparing these conditions, one can observe that the amount of free chain was clearly increased in the absence of aggregates. Likewise, in the *ltn1∆*background (i.e., with endogenous Rqc2 present) overexpression of Rqc2 clearly led to more quantitative conversion of reporter monomers into both CATylated and aggregated forms (Figure 2).

*Reviewer #3: Ribosome-associated quality control (RQC) is an emerging topic with lots of surprises to be discovered. In this study, Yonashiro et al. reported that Rqc2 not only mediates the tagging of nascent chains with AT sequences, but also promotes their aggregation if the tagged nascent chains are not ubiquitinated or degraded. While the former finding was reported by Shen et al.*

*(Science 347, 2015), the aggregation feature reported by this study is interesting and significant. In particular, it might shed light on the neurodegenerative phenotypes of mice lacking Ltn1. The enthusiasm about the aggregation feature of RQC is reduced a bit by some incomplete results interpretation. The role of chaperone Sis1 in this process, although interesting, remains obscure. With more supporting evidence from additional experiments, I would support publication of this exciting story in eLife. Major concerns:*

*1) The authors nicely used many different approaches to prevent high molecular weight aggregates formation. For instance, addition of poly-Lys (Figure 1) and depleting Rqc2 (Figure 2) abolished the aggregates formation. But why the free products (NS/K12) were not increased in a corresponding manner? Where were the non-aggregated products?*

Please see response under “Essential revisions” above.

*2) In Figure 2, cells lacking Ltn1 apparently had more aggregates then cells lacking Rqc1. Does this mean ubiquitination could play a role in reducing aggregates formation? Is it possible to show the ubiquitin levels of the reporter (NS/K12) in these two strains?*

We have so far not noticed a consistent trend regarding the relative levels of aggregates formed in Ltn1-deficient versus Rqc1-deficient cells. See, for example, Figure 3 – the signal from endogenous aggregates detected via Sis1 immunoblot appears to be more intense in Rqc1-null cells. Although the reviewer raises an interesting possibility, in the short time available for review it was not possible to address this point experimentally. We hope the reviewer agrees that this point is beyond the scope of the manuscript.

*3) In Figure 2, it is clear that only the sequence of AT10 is able to trigger the aggregation of the GFP reporter. But in WT cells, AT10-tagged GFP is evidently resistant to degradation. Is there any explanation about this discrepancy?*

We respectfully disagree with the reviewer that there is a discrepancy in the data. The AT10 reporter does have stop codons and has no stalling sequence, so it is expected to terminate translation normally; it aggregates because of Ala-Thr repeats hard-coded in the reporter sequence. Given that AT10 is not an RQC substrate, its levels are not expected to be different in WT cells in comparison to RQC-deficient cells. We have revised the text to clarify these points.

*4) The authors provide strong evidence that aggregates of the nascent chain is due to the "CAT" tailing. But it is still suggestive at most. It would be ideal to demonstrate that the aggregates indeed contain the unique AT sequence. Mass spectrometry might be possible in this case.*

We are pleased with the reviewer’s statement that we “provide strong evidence that aggregates of the nascent chain is due to the "CAT" tailing.” We agree it would be nice to show that the aggregates indeed contain the unique AT sequence. Examining this is feasible, but would take long to address – we presume that the mass spec. approach may turn out to be challenging because CAT tails are heterogeneous and low-complexity sequences.

*5) The presence of Sis1 in the aggregates is intriguing. But what is the role of Sis1 in this process? Sis1 has been reported to influence prion formation (Harris et al. PLoS Genet 2014). What happens to the RQC aggregate formation in cells lacking Sis1?*

Please see response under “Essential revisions” above.

6) In Figure 4, the authors pointed out K-less R12 reporter could be modified but failed to form aggregates. Is Sis1 still interacting with these species? This is important information that could explain the different behavior of this reporter in wild type cells.

We presume that the inability to detect K-less R12 aggregates was due to a detection issue, given that levels of the GFP K-less R12 reporter were characteristically lower compared to the GFP-R12 reporter.

Properly addressing the reviewer’s question is also made difficult by this difference in expression levels among the various reporters. Nonetheless, using a different reporter set (based on the Protein A ZZ domain-Ribozyme fusion) we now show that the PtnA-Rz K-less reporter could both be CATylated and form aggregates in wild type cells, so stalled K-less proteins do not appear to have a problem aggregating in general.